# Hormonal and Genetic Regulatory Events in Breast Cancer and Its Therapeutics: Importance of the Steroidogenic Acute Regulatory Protein

**DOI:** 10.3390/biomedicines10061313

**Published:** 2022-06-03

**Authors:** Pulak R. Manna, Ahsen U. Ahmed, Deborah Molehin, Madhusudhanan Narasimhan, Kevin Pruitt, P. Hemachandra Reddy

**Affiliations:** 1Department of Internal Medicine, Texas Tech University Health Sciences Center, Lubbock, TX 79430, USA; hemachandra.reddy@ttuhsc.edu; 2Comprehensive Cancer Center, University of California Davis, Sacramento, CA 95817, USA; ahsen.ahmed@ttuhsc.edu; 3Immunology and Molecular Microbiology, Texas Tech University Health Sciences Center, Lubbock, TX 79430, USA; deborah.molehin@ttuhsc.edu (D.M.); kevin.pruitt@ttuhsc.edu (K.P.); 4Neuroscience and Pharmacology, Texas Tech University Health Sciences Center, Lubbock, TX 79430, USA; madhusudhanan.narasimhan@utsouthwestern.edu; 5Neurology, Departments of School of Medicine, Texas Tech University Health Sciences Center, Lubbock, TX 79430, USA; 6Public Health Department of Graduate School of Biomedical Sciences, Texas Tech University Health Sciences Center, Lubbock, TX 79430, USA; 7Department of Speech, Language and Hearing Sciences, School Health Professions, Texas Tech University Health Sciences Center, Lubbock, TX 79430, USA

**Keywords:** breast cancers, StAR, estrogen, E2 biosynthesis, hormone receptors, HDAC inhibitors, therapies

## Abstract

Estrogen promotes the development and survival of the majority of breast cancers (BCs). Aromatase is the rate-limiting enzyme in estrogen biosynthesis, and it is immensely expressed in both cancerous and non-cancerous breast tissues. Endocrine therapy based on estrogen blockade, by aromatase inhibitors, has been the mainstay of BC treatment in post-menopausal women; however, resistance to hormone therapy is the leading cause of cancer death. An improved understanding of the molecular underpinnings is the key to develop therapeutic strategies for countering the most prevalent hormone receptor positive BCs. Of note, cholesterol is the precursor of all steroid hormones that are synthesized in a variety of tissues and play crucial roles in diverse processes, ranging from organogenesis to homeostasis to carcinogenesis. The rate-limiting step in steroid biosynthesis is the transport of cholesterol from the outer to the inner mitochondrial membrane, a process that is primarily mediated by the steroidogenic acute regulatory (StAR) protein. Advances in genomic and proteomic technologies have revealed a dynamic link between histone deacetylases (HDACs) and StAR, aromatase, and estrogen regulation. We were the first to report that StAR is abundantly expressed, along with large amounts of 17β-estradiol (E2), in hormone-dependent, but not hormone-independent, BCs, in which StAR was also identified as a novel acetylated protein. Our in-silico analyses of The Cancer Genome Atlas (TCGA) datasets, for StAR and steroidogenic enzyme genes, revealed an inverse correlation between the amplification of the *StAR* gene and the poor survival of BC patients. Additionally, we reported that a number of HDAC inhibitors, by altering StAR acetylation patterns, repress E2 synthesis in hormone-sensitive BC cells. This review highlights the current understanding of molecular pathogenesis of BCs, especially for luminal subtypes, and their therapeutics, underlining that StAR could serve not only as a prognostic marker, but also as a therapeutic candidate, in the prevention and treatment of this life-threatening disease.

## 1. Introduction

An overwhelming amount of evidence indicates that estrogen and its receptors play crucial roles in the pathogenesis and progression of breast cancer (BC), which is the second most prevalent malignant disorder in women and accounts for ~26% of all cancer cases, and is the leading cause of mortality globally. The biological actions of estrogens are mediated by binding to estrogen receptors, ERs: ERα and ERβ (encoded by *ESR1* and *ESR2* gene, respectively), which belong to the nuclear receptor subfamily of ligand-activated transcription factors. ERs form either homo- or hetero-dimers and bind to the estrogen response element (ERE) present at the promoter region of target genes and regulate transcription [1,2]. It is noteworthy that estrogen-dependent malignant breast tumors largely express ERα that essentially transduces the activities of estrogens, which are synthesized from androgens by action of the aromatase enzyme [3]. Aromatase (also known as estrogen synthetase/synthase), a member of the cytochrome P450 superfamily, is localized at chromosome 15 in the human genome [4,5]. A single gene encodes aromatase (*CYP19A1*) in humans, and it is expressed in a variety of tissues, including ovary, testis, placenta, bone, skin, brain, and adipose tissue. Epidemiological and experimental evidence indicate that malignant breast tissues express high levels of aromatase, along with large amounts of estrogens, especially E2 [5,6]. Therefore, aromatase is a molecular target for therapeutic approaches for BCs, and other relevant malignant disorders [7,8]. Elevated levels of E2 in stage specific BCs have been demonstrated to be 10 to 50 times higher than those of their normal counterpart and/or in circulation [9,10]. The majority of BCs (~80%) are hormone sensitive and express ER (ERα), progesterone receptor (PR), and human epidermal growth factor receptor 2/the erythroblastosis oncogene-B2 (HER2/ErbB2) [1,11]. The remaining 20% BCs that do not express ER, PR, and HER2 are termed as triple negative BCs (TNBCs) [12]. The proliferative and DNA-damaging effects of estrogen and its metabolites are known to be linked with the aberrant growth and transformation of hormone-sensitive BCs [9,13]. Importantly, TCGA research datasets provide comprehensive information to understand the molecular factors involved in various breast malignancies and, consequently, their prognosis and therapeutic potentials [14,15,16].

The regulation of steroid biosynthesis is primarily mediated by the StAR protein (also known as STARD1, or StAR-related lipid transfer domain-containing protein 1), whose expression, activation and extinction are mediated by a host of signaling pathways [17,18,19,20]. In addition, there is a wealth of information that agents and/or factors that influence StAR expression also influence steroid biosynthesis through endocrine, autocrine, and paracrine mechanisms in classical and non-classical tissues. Considering the importance of StAR in the transport of cholesterol, as well as steroid biosynthesis, its appropriate regulation is the key to proper functioning of a variety of cholesterol and/or steroid led processes. Conversely, the dysregulation of the steroidogenic machinery, involving androgen and estrogen biosynthesis, has been associated with the pathogenesis of a variety of hormone-sensitive cancers, including BCs [2,8,21,22]. While StAR plays an indispensable role in the regulation of steroid biosynthesis, its involvement in BCs remains poorly understood. Conspicuously, we reported that StAR is overexpressed in hormone-responsive BC cells (in comparison with little to none in non-cancerous mammary epithelial cells), in which amplification of the *StAR* gene is correlated with poor survival of patients afflicted with this deadly disease [23,24].

Since estrogen plays a central role in the pathogenesis of BCs, therapeutic strategies targeting either aromatase or ERs have evolved [5,9,10]. In fact, a number of aromatase inhibitors (AIs) are routinely used for the treatment and management of hormone-responsive BCs in post-menopausal women, however, they decrease estrogen synthesis throughout the body. Thus, AIs generate adverse side effects since estrogens play important roles in many physiological processes, such as bone mineralization, cognitive functions, lipid and carbohydrate metabolism, and vascular biology [10]. The undesirable effects of AI therapy have led to the development of newer treatment options that are targeted with little to no side effects. Accordingly, recent studies have demonstrated a link between HDACs and StAR, aromatase, and estrogen biosynthesis [2,21,23,25]. The HDAC family consists of 18 members that are divided into four classes based on their sequence similarity and functional specificity. While HDAC classes I (HDACs 1, 2, 3, and 8), II (HDACs 4-7, 9, and 10), and IV (HDAC 11) are Zn^2+^ dependent, class III (Sirtuin family; Sirt1-7) requires NAD^+^ for catalytic activity [2]. Interestingly, HDAC inhibitors (HDACIs), possessing anti-tumorigenic properties, have been reported to exhibit favorable outcomes for the management of a variety of diseases, including BCs [26,27,28,29]. Herein, we discuss the current understanding of the pathogenesis of BCs with respect to hormonal and genetic regulatory events, as well as potential therapeutic approaches for the treatment of this devastating disease. 

## 2. Overview of BC Subtypes

BCs are frequently classified based on immunohistochemical (IHC) markers for ER (specifically ERα, but not ERβ), PR, and HER2 into the following subtypes: luminal A, luminal B, HER2, normal-like, and TNBC (in which 15–20% are basal-like) (Figure 1). While luminal A tumors are ER positive (ER+) and/or PR+, and do not express HER2 (HER2−), luminal B subtypes essentially express ER+/PR+ and HER+ [2,24,30]. Notably, luminal categories are hormone sensitive and account for over three quarters of all BCs. Additionally, HER2 subtypes are ER−, PR−, and HER+, and TNBC tumors do not express ER−/PR− and HER−. Both HER2 and TNBC subtype tumors are generally more aggressive with advanced presentation at diagnosis and worse outcomes. Both histological and clinical analyses are commonly used for characterization and staging of various BC subtypes. The pathologic staging for BCs is the TNM (tumor, node, and metastasis) system, ranging from stages 0 to IV, in which 0 and IV represent early and advanced and/or metastatic (spreading to distant organs) stages, respectively [31]. 

## 3. Molecular Pathogenesis of BCs and Frequently Altered Pathways

Hormonal, genetic, environmental, and reproductive factors contribute to the development and progression of BCs. Nonetheless, BC is a multifactorial condition with diverse biological and histopathological features, and is characterized as several neoplastic disorders, each with a distinct molecular profile that ultimately determines disease course and the tumor’s cellular response towards treatment. A vast majority of BCs are ER+ and responsive to estrogens in promoting tumor proliferation. Estrogens produced in either extra-ovarian sites, locally within malignant breast epithelial cells, or circulation are the key drivers of BCs in post-menopausal women via paracrine and/or intracrine mechanisms [32,33]. Regardless of sources, it is unambiguous that an exposure of estrogens leads to the expression of stem cell markers and amplified oncogenic signaling towards increased susceptibility to BCs [34,35]. Even so, a number of BC risk factors include menopause, age, family history, early menarche, obesity, oral contraceptives, and hormonal treatment during menopause (reviewed in Refs. [2,8]). Hormones influencing reproductive function are thought to induce BC risk by promoting cell proliferation and growth, accompanied by DNA damage and genomic instability [10,36]. However, it is worth noting that estrogens are known to induce apoptosis in BC cells and are capable to overcome endocrine therapy resistance [37,38]. 

Multiple signaling pathways modulate various cellular processes, such as protein synthesis, metabolism, growth, cell cycle progression, angiogenesis, and apoptosis, and the dysregulation of these events plays a vital role in the pathogenesis of BCs [39]. Genetic aberrations in malignant cells play a central role in BC development, and have been analyzed with DNA copy number alterations, changes in gene expression, and mutations [40,41]. TCGA, in coordination with the National Cancer Institute’s Center for Cancer Genomics and the National Human Genome Research Institute, offers comprehensive datasets to accelerate understanding of the molecular basis of cancers though genome sequencing and bioinformatics [14,15,16]. Analyses of TCGA BC datasets identified several mutations in both previously implicated genes (*PIK3CA,* phosphatase and tensin homolog (*PTEN*), *AKT1*, tumor protein p53 (*TP53*), *GATA3*, *CDH1*, retinoblastoma 1 (*RB1*), *MLL3*, *MAP3K1*, and *CDKN1b*) and new genes (*TBX3*, *RUNX1*, *CBFB, AFF2*, *PIK3R1*, *PTPN22*, *PTPRD*, *NF1*, *SF3B1*, and *CCND3*) [14,15,16]. Table 1 illustrates the expression and mutation of hormone receptor (HR) genes identified by the TCGA study for BC molecular subtypes associated with clinical and pathological features.

The canonical pathways altered in BCs include phosphoinositide 3-kinase (PI3K)/PTEN, TP53, RB, and Jun N-terminal kinase (JNK)/p38 mitogen activated protein (MAPK) (Figure 2). Mutated genes are more recurrent and diverse in luminal BC subtypes, even though the rate of mutation is higher in HER2E and basal-like categories [14]. Estrogen has been reported to influence the PI3/AKT and steroid receptor coactivator (Src)/MAPK pathways that modulate proliferation of MCF7 BC cells [42]. Studies have shown that crosstalk signaling between ER, RAS/RAF/MAPK and PI3K/AKT results in conformational changes towards the activation of various processes, ranging from cellular differentiation to mitogenesis to apoptosis [43,44]. 

The PI3K signaling pathway is recurrently altered in BCs [39]. PI3Ks represent a family of heterodimeric kinases, which consist of a catalytic subunit known as p110 with four isoforms (encoded by *PIK3CA*, *PIK3CB*, *PIK3CD*, and *PIK3CG*) and a regulatory subunit (isomers p85α, p85β, p55γ encoded by *PIK3R1*, *PIK3R2*, and *PIK3R3*), respectively. While the class IA PI3Ks are activated by receptor tyrosine kinases, including EGFR, insulin-like growth factor 1 receptor, and fibroblast growth factor receptor (FGFR), the class IB is stimulated by G-protein coupled receptors. The class I PI3Ks phosphorylate phosphatidylinositol 4,5-bisphosphate resulting in phosphatidylinositol 3,4,5-trisphosphate that activates AKT signaling [45]. The activated AKT, in turn, interacts with a wide range of substrates, including the mammalian target of rapamycin, mouse double minute 2 homolog, nuclear factor of κB, and cyclin-dependent kinase 4 (CDK4), and triggers downstream signaling pathways (Figure 2). The PI3K/AKT pathway regulates cell growth, survival, and evasion of apoptosis, and exerts an inverse relationship with its negative regulator, *PTEN*, a tumor suppressor gene [45,46,47]. 

RAS is a family of GTPases that are activated mainly through the action of external ligands on receptor tyrosine kinases, which are composed of RAS, RAF, MEK/MAPKK, and MAPK. RAS triggers the protein kinase activity of RAF, which phosphorylates MEK/MAPKK and regulates the transactivation potential of genes involved in cell cycle regulation [14,48]. Abnormal activity of the RAS-RAF-MEK-MAPK signaling cascade plays a vital role in carcinogenesis by influencing signals that enhance cell proliferation and angiogenesis, and down-regulate apoptosis. Inactivating mutations in the *TP53* tumor suppressor gene contribute to development of a variety of cancers. These mutations include MDM2 amplification and loss of ataxia telangiectasia-mutated (*ATM*) gene, encoding a protein involved in DNA repair, and both occurred commonly in luminal B tumors, indicating that the TP53 pathway remained largely intact in the majority of malignant tumors. Conversely, TP53 mutations were present in 75% of HER2E and 84% of basal-like tumors. The RB1 tumor suppressor gene pathway revealed that luminal A tumors with the best prognosis retain the activity of RB1 while more aggressive tumors, such as TNBC and/or basal-like categories, were likely to lose RB1 [14,49]. Therefore, the understanding of various signaling pathways altered in various BC subtypes and the availability of genomic and clinical data may permit more targeted therapy in the management of BCs. 

## 4. Estrogen and Its Receptors in BCs

It is well established that estrogen and its receptors are involved in the etiology of BCs. As stated above, estrogens exert their effects by binding to two ERs, ERα (*ESR1*) and ERβ (*ESR2*), which possess high sequence homology, share many properties, form dimers, and bind to the ERE in regulating transcription. Accumulating evidence indicates that ER can bind DNA directly, or indirectly by tethering transcription factors which then recruit coregulators in influencing chromatin remodeling, as well as transcriptional machinery [50,51]. The transcriptional regulation of ER is a complex process that involves the simultaneous interaction of various coregulatory factors and crosstalk between diverse signaling pathways. Studies have shown that ERs, especially ERα, interact with a number of transcription factors, including activator protein 1, nuclear factor-kB, and specific protein 1, and regulate target gene transcription [52,53]. Transcriptional synergy of ERs requires the interaction of multiple transcription factors and their post-translational modifications, along with coregulator molecules, revealing either coactivator or corepressor effects [53,54]. Coregulators, acting as bridging proteins, function in either switching on (activation) or silence (repression) ER gene transcription in a variety of conditions. Whereas coactivators, such as SRC/p160, CREB-binding protein (CBP/)/p300, and E3 ubiquitin protein ligases, by interacting with ligand-binding domains of ER, enhance transcription, corepressors, such as NCoR and SMRT, by recruiting HDACs, inhibit ER-mediated gene transcription by stabilizing chromatin [52,55,56]. These findings point to the role played by transcription factors, coregulators, and chromatin modifiers in the regulation of ER expression and their downstream target gene transcription in impacting breast physiology and pathophysiology. 

A number of compounds with affinities to ERs have been reported to act as both agonists and antagonists, and behave as estrogens and antiestrogens, in tissue-specific manners, and are termed as selective ER modulators (SERMs) [57,58], which exhibit a variety of effects through ER-dependent and ER-independent signaling. SERMs, especially the most frequently used tamoxifen for the treatment of BCs in post-menopausal women, antagonize the actions of estrogens, thus, block hormone-responsive proliferation and maintenance of BCs [58,59]. It is noteworthy that cholesterol, the obligate precursor of all steroid hormones, including estrogens/E2, and its metabolites, plays a pivotal role in BC pathophysiology [60,61,62]. In accordance, studies have shown that the cholesterol metabolite 27-hydroxycholesterol (27-HC), by interacting with ER, provokes hormone-dependent BCs [60,63]. These reports are reminiscent of our findings that demonstrated that expression of the critical cholesterol transporter, StAR, along with E2 accumulation, is aberrantly high in ER+/PR+ BCs [23,24]. Moreover, the involvement of the 5,6-epoxycholesterol (5,6-EC) metabolic pathway is evident in BCs, in which an oncometabolite 6-oxo-cholestan-3β,5α-diol (OCDO), derived from 5,6-EC, has been shown to drive breast tumorigenesis through the glucocorticoid receptor, a process that involves the liver X receptor (LXR) cascade, but not ERα signaling [64]. Concomitantly, levels of OCDO, including the enzymes involved in its production, have been demonstrated to be higher in patients’ breast tumors in comparison to their non-cancerous counterparts [65]. It is unequivocal that oncogenic signaling in the breast is influenced by a plethora of processes, including cholesterol and its diverse metabolites, and SERMs modulate a number of these factors/signaling for combating BCs. Regardless of the mechanism involved, there is increasing evidence that SERMs and/or tamoxifen, by inhibiting 5,6-EC metabolites, enzymes in the cholesterol biosynthetic pathway, and LXR regulatory events, prevent hormone-sensitive BCs [65,66,67,68]. Based on the above considerations, it is conceivable that SERMs affect StAR, or other relevant cholesterol transporters, such as STARD3, in ER+/PR+ BCs, and require future investigations.

Estrogen signaling is particularly triggered or obliterated depending upon an equilibrium between ERα and ERβ in specific tissues. Notably, estrogen-dependent proliferation and the growth of BCs are mediated by ERα [69]. Alternatively, ERβ is reported to act as a tumor suppressor [70]. Both ERα and ERβ co-express in a number of benign and malignant breast tissues; however, the expression of ERα is higher than that of ERβ in hormone-sensitive BCs [3,71]. ER expression is influenced by microRNAs (miRNAs), either inhibiting its translation, triggering post-transcriptional degradation, or repressing ER-associated signaling. MiRNAs control genes involved in many cellular events, including cell cycle regulation, cell differentiation, programmed cell death, and migration, and have been implicated in many pathophysiological processes, such as tumor initiation, invasion, and metastasis [72,73]. Following estrogen treatment, miRNA profiles of miR-206, miR-125a/b, miR-17-5p, miR-34a, and Let-7 members with tumor suppressive abilities are dysregulated, whereas miR-21, miR-155, and miR-10b may be associated with oncogenic potentials in BCs [74]. The overexpression of miR-206 in MCF7 cells decreases ER, as well as endogenous levels of their target genes, coactivators, and transcription factors, including cyclin D1, PR, pS2, Src-1/3, and GATA-3. Hedgehog (Hh) signaling has been implicated in embryogenesis and when aberrantly activated results in progression and sustenance of breast tumors [75]. Higher levels of Hh effector protein glioblastoma 1 (GLi1) have been reported in tamoxifen-resistant BC cells. The depletion of GLi1 reduces ER expression in both tamoxifen-resistant and responsive BCs [76], leading to a reduction in ERα activity and eventually culminating in a decline in cell proliferation. 

## 5. Functional Relevance of StAR and Steroidogenic Machinery in BCs

The transport of intramitochondrial cholesterol by the StAR protein represents a key step in the regulation of steroid biosynthesis, in which the cytochrome P450 cholesterol side chain cleavage enzyme converts cholesterol into the first steroid, pregnenolone, which is then metabolized to various steroids by a series of enzymes in pertinent tissues (Figure 3). 

Malfunction in the steroidogenic pathway, involving androgen and/or estrogen biosynthesis, has been implicated in the pathogenesis of a number of cancers, including the most prevalent hormone-sensitive BCs [2,24,77]. We reported that ER+/PR+ BCs exert a markedly higher expression of StAR, along with increased E2, and their levels are little to none in either normal mammary epithelial cells or TNBCs [23]. In accordance with this, analyses of TCGA datasets demonstrated that the amplification of the *StAR* gene, but not other steroidogenic enzyme genes, is correlated with poor survival of BCs [24]. Hence, aberrant high expression of StAR presumably facilitates abnormal delivery of cholesterol to the mitochondrial inner membrane, which results in adequate availability of cholesterol for overproduction of E2 in promoting breast tumorigenesis. Consequently, StAR-mediated delivery of cholesterol, following substantial increase in estrogen/E2 production, could be a plausible mechanism in the development and growth of luminal subtype BCs. Even so, the role of STARD3, a late endosomal membrane protein with structural and functional homology to StAR, in the growth and maintenance of hormone-sensitive BCs, cannot be excluded. Studies have shown that STARD3 is overexpressed in HER2+ BCs, with its gene located on the HER2/ERBB2-containing gene locus [2,24].

Aromatase (*CYP19A1*) catalyzes the final step in estrogen biosynthesis. The regulation of the *CYP19A1* gene is mediated by 10 alternatively spliced promoters in normal and malignant breast tissues, in which hormones, cytokines, growth factors, as well as a host of signaling pathways, play permissive roles [2,5,36,78]. While post-translational modification (PTM) of aromatase by phosphorylation has been established, we reported the importance of a novel PTM, acetylation, in aromatase regulation [21]. Regardless of various PTMs, aberrant high intra-tumoral expression of aromatase critically influences the development and maintenance of ER+/PR+ BCs. Consequently, the accumulation of E2 has been demonstrated to be strikingly higher in malignant breast tissues compared to non-malignant mammary epithelial cells [2,23,79]. In pre-menopausal women, estrogens are synthesized by the ovarian granulosa and placental corpus luteal cells, which reaches the malignant tissue through systemic circulation [2,78]. However, at menopause estrogen is mainly produced in extra-ovarian tissues, such as adipose tissue, brain, bone, and skin. While estrogen biosynthesis in plasma declines in post-menopausal women, androgen level remains relatively unchanged for many years. Androgens are the substrate for estrogens in peripheral tissues (Figure 3), thus, post-menopausal women are more susceptible to developing BCs [5,9,10]. In fact, estrogens, derived from a number of sources, are the major drivers of hormone-sensitive BCs, although several factors are instrumental in the survival and maintenance of this devastating disease. 

## 6. Steroidogenic Enzyme and HR Genes, and Their Correlation to BC Survival

High-throughput sequencing TCGA datasets have been helpful in understanding the molecular factors involved in the pathogenesis of BCs, their diagnosis, and therapeutic strategies [14,15,16]. Previously, we analyzed TCGA genomic datasets and reported that the majority of BCs belong to luminal subtypes that express ER (77%), PR (64%), and HER2 (15%) [2,24]. In addition, whereas the expression of steroidogenic enzyme genes (*CYP11A1*, *HSD3B*, *CYP17A1*, *CYP19A1*, and *HSD17B*) was altered at varying levels, the amplification of the *StAR* gene (observed 12–26% in different TCGA studies/publications) was correlated with poor survival of BCs [24]. Moreover, TNM analyses of StAR mRNA revealed BC deaths in stage-dependent manners. In line with these findings, we demonstrated that expression of StAR is markedly higher in hormone-dependent, but not hormone-independent, BCs [23], suggesting that StAR, by providing the precursor of estrogens, acts as a tumor promoter in ER+/PR+ BCs.

The impact of HRs in BC survival was evaluated in conjunction with METABRIC (Molecular Taxonomy of Breast Cancer International Consortium; 2173 tumors) genomic datasets [14,15,16], utilizing DNA copy number alterations that were categorized as diploid (normal/no change), gain (+1 copy), amplification (+2 copies), hemizygous deletion (−1 copy), and homozygous deletion (−2 copies) [2,24]. Specifically, the amplification of various HR genes, i.e., *ESR1*, *ESR2*, *PGR*, and *ERBB2,* and StAR as well, was analyzed for their correlation with the survival of BCs by generating Kaplan-Meier plots utilizing ‘diploid’ and ‘amplification’ categories (Figure 4). It can be seen that amplification of both *ESR1* and *ESR2* genes was inversely correlated with the survival of BC patients (panel A), in which the median survival rate was noticeably decreased in both cases, reinforcing the notion that ERs play central roles in the progression and maintenance of estrogen sensitive BCs. Whereas amplification of the *PGR* gene (panel B) was not found to be associated with BC survival, *ERBB2* gene expression was strongly correlated with poor survival of BC patients (panel C). The latter strengthen the concept that BC tumors expressing HER2+ are more aggressive with worst prognoses, and do not respond to endocrine therapies. Interestingly, the amplification of the *StAR* gene (panel D) was significantly correlated with poor survival of BC patients, and these data are in support of our previous findings [23], providing novel insight that StAR plays a key role in the pathogenesis of ER+/PR+ BCs. 

METABRIC datasets were analyzed for a number of genes that are known to be commonly amplified in BCs. As depicted in Figure 5, several genes amplified between 22% and 0.2% are the following: MYC (22%), *MDM4* (22%), *PARP1* (20%), *PTK2* (18%), *E2F5* (17%), *RAB25* (17%), *MMP16* (17%), *HEY1* (16%), *CCND1* (14%), *ERBB2* (14%), *StAR* (14%), *FGFR1* (12%), *EIF4EBP1* (12%), *RPS6KB1* (9%), *MAP3K3* (7%), *PDPK1* (6%), *MLST8* (6%), *MMP25* (6%), *RPS6KB2* (4%), *RBPJL* (4%), *RFNG* (4%), *TERC* (4%), *TERT* (4%), *PIK3CA* (4%), *RPTOR* (4%), *CDKN2A* (3%), *KDM5A* (3%), *EIF5A2* (3%), *PTEN* (2.2%), *MAP2K4* (2%), *KMT2C* (1.4%), *RB1* (1%), and *TP53* (0.2%). These genes are chosen arbitrarily and their amplifications in survival of BCs are not investigated in this study. Interestingly, the amplification of the *StAR* gene in poor survival of BCs was corroborated with the fact that STARD3 was initially identified as an amplified gene in BCs [80]. It is plausible that the overexpression of StAR, resulting in excess delivery of cholesterol along with a substantial increase in estrogen/E2 synthesis, is a fundamental event in the progression and maintenance of hormone-sensitive BCs, which opens up a new avenue in BC research. 

## 7. Putative Roles of Multiple Factors/Signaling in StAR Linked BCs

The StAR protein, by controlling the transport of intramitochondrial cholesterol, mediates the biosynthesis of steroid hormones, which play essential roles in diverse processes [18,20]. Multiple levels of regulation impinge on StAR function; thus, it is plausible that a host of extracellular factors and/or signaling pathways putatively link to the development of StAR-driven, hormone-responsive BCs (Figure 6). 

### 7.1. Cholesterol and Sex Steroids

Cholesterol utilized for steroidogenesis is derived from a number of sources, i.e., de novo synthesis of cellular cholesterol, lipoprotein-derived cholesteryl esters, and hydrolysis of cholesteryl esters stored in lipid droplets. Regardless of sources, the conversion of cholesteryl esters into free cholesterol is an important step in controlling steroidogenesis. We demonstrated that hormone-sensitive lipase (HSL), a neutral cholesteryl ester hydrolase, plays a pivotal role in regulating intracellular cholesterol metabolism, and this process is instrumental in StAR-mediated, steroid led events [20,81]. In accordance with this, the regulation of either estrogen/E2 or progesterone has been shown to activate BC growth and maintenance [2,21,82]. It has been reported that inclusion of progesterone during endocrine therapy led to the development of BC tumors in a clinical trial of 16,608 patients [83]. Of note, several pathways, including EGFR, cAMP/PKA, and MAPK, which play central roles in the survival and growth of breast tumors, are activated via estrogen signaling [84]. These findings are in support of our data that demonstrated that abnormally high StAR and E2 levels in ER+/PR+ BC cells promote breast tumorigenesis [23,24]. 

### 7.2. Aldosterone and Fibrogenic Signaling

The upregulation of StAR has been shown to stimulate aldosterone-mediated pulmonary remodeling and cause vascular fibrosis [85]. Excessive aldosterone accumulation also causes inflammation, endothelial dysfunction, and alter metabolic sequelae [86], which are connected to BC development [87,88,89]. Aldosterone has been reported to contribute to the growth and migration of BC cells by inducing an interaction between mineralocorticoid receptor and G-protein-coupled estrogen receptor (GPER), as well as EGFR [90]. As such, as a driver of aldosterone/fibrogenic signaling, StAR may promote hormone-dependent BCs by utilizing vascular pathways.

### 7.3. Reproductive Hormone, Behavior, and Stress Changes

Steroid hormones influence normal reproductive function, bodily homeostasis, and response to stress, in which steroid biosynthesis is governed by the mitochondrial StAR protein. It is worth noting that StAR expression is tightly correlated with the synthesis of steroids in a variety of steroidogenic tissues [20,91,92]. In the central nervous system, steroid hormones exert their effects on various neurotransmitters, such as GABA, serotonin, dopamine, and glutamate, which regulate diverse functions ranging from mood, behavior, cognitive abilities, and stress responses [93]. It is recognized that fertility drugs, e.g., estrogen and progesterone, increase the risk of developing BCs [94,95]. Additionally, the overproduction of steroid hormones resulting in chronicity of stress system activation is well known to have a negative behavioral and stress response, leading to neurobiological hypersensitivity and impacting immune, inflammatory, and metabolic disorders [2,96]. All of these processes, connecting StAR and steroid biosynthesis, contribute to the progression and maintenance of BCs.

### 7.4. Toll-Like Receptors and Immuno-Inflammation

The overexpression of StAR has been shown to affect toll-like receptor 3 (TLR3), TLR6, and lymphotoxin alpha (LTα) activities, which contribute to various cancers [97]. TLR3 is considered to play a vital role in the recognition of endogenous macromolecules released by injured tissue, in which double-stranded viral and retroviral RNAs influence the initiation and continuation of inflammatory and immune responses [98]. TLR3 stimulation has been reported to promote BC cells toward a cancer stem cell phenotype and, notably, an increased TLR3 expression in BC patients is shown to have a poor prognosis [99]. Lymphotoxin-α/β (LTα/β) are pro-inflammatory related cytokines of the tumor necrosis factor superfamily, and an LTα/β-driven link between immune-inflammation and cancer via changes in tumor microenvironment has been reported in BCs [100]. Thus, by increasing the factors that play a key role in the immune-inflammatory cancer network, StAR can promote progression of ER+/PR+ BCs.

### 7.5. Oxysterols

Oxysterols (oxygenated derivatives of cholesterol), specifically 27-HC, a product of *CYP27A1* gene, are known to influence cholesterol metabolism and regulation. Oxysterols are ligands for ERs and LXRs [60,101]. Accumulating evidence indicates that 27-HC plays a vital role in the development of BCs. Studies have identified a higher concentration of 27-HC in ER+ breast tumors, in which 27-HC exerts an estrogen-mimetic effect in promoting tumorigenesis [60,102]. In the absence of E2, 27-HC is shown to act as an agonist for ERα to enhance cell adhesion and modulate pro-inflammatory events [103], which are supportive for tumor cells to sustain their proliferative activities [104]. It has been demonstrated that StAR enhances 27-HC levels, and the overexpression of CYP27A1 is capable of increasing StAR in HepG2 liver cells [105]. Human endometrial stromal and granulosa cells treated with oxysterols, including 27-HC, have been shown to elevate StAR and steroid levels [106]. We reported that HSL-mediated StAR expression and steroid biosynthesis entails enhanced oxysterol production in controlling a variety of steroid-led activities [20,81]. Thus, it is likely that an abundance of 27-HC-StAR-steroid levels could act in a cyclical manner and may play a pro-carcinogenic role in hormone-sensitive BCs (Figure 6). 

### 7.6. Mitochondrial Cholesterol

The mechanism underlying mitochondrial cholesterol accumulation in cancer cells is poorly understood, even though StAR regulates the transport of intra-mitochondrial cholesterol. Noteworthy, however, dynamin-related protein-1 (DRP1) is a regulator of mitochondrial fission and it is upregulated in BCs compared with normal mammary tissues [107]. Furthermore, the overexpression of DRP1 is correlated with poor survival of BC patients. A pioneering study uncovered that the expression and activity of StAR contribute to mitochondrial intermembrane trafficking of cholesterol in hepatocellular carcinoma [108]. Importantly, the StAR loss of function, enhancing chemotherapy susceptibility, showed that enrichment of StAR-mediated mitochondrial cholesterol can be responsible for chemotherapy resistance. Even so, elevated cholesterol accumulation has been associated with BC risk, as it is a precursor for estrogen and progesterone, which are known to contribute to development and maintenance BCs [109,110]. 

### 7.7. Bile Acid and Lipid Metabolism

The regulation of cellular cholesterol and lipid homeostasis is influenced by bile acid biosynthetic pathways, in which the acidic/alternate pathway plays a predominant role [111]. By interacting with farnesoid X receptor and binding to a G-protein coupled cellular receptor, TGR5, bile acids can also act as signaling molecules to control carbohydrate metabolism [112]. StAR has been shown to increase the activity of *CYP27A1*, production of oxysterols, and rates of bile acid synthesis [20,81]. An earlier connection between bile acid and BC progression was drawn by a study that showed that bile acids exert estrogen-like effects, thus, increasing the development of BCs [111]. In addition, a correlation between bile acid and estrogen sensitive BCs has been reported by a clinical study that showed ~52% of post-menopausal women suffering with BCs exhibited elevated levels of plasma bile acids in relation to the healthy controls [113]. As such, StAR, by increasing bile acid-associated changes, may promote BC progression.

### 7.8. LON, SPG7, and AFG3L2

The ectopic expression of StAR upregulates the transcription of mitochondrial proteases, such as Lon peptidase1 (LON), Spastic Paraplegia (SPG7), and ATPase Family Member3-Like2 (AFG3L2). This heterologous overexpression model-based outcome of transcriptional activation is termed ‘StAR overload response,’ which facilitates the degradation/removal of excess StAR in maintaining mitochondrial protein quality control system [114,115]. LON, SPG7, and AFG3L2 are ubiquitous proteases embedded in the mitochondrial membrane that governs the protein quality surveillance processes by removing excess, misfolded, modified, or damaged proteins, thereby controlling the stability and functional integrity of the mitochondrial genome [116]. In general, the disruption of proteostatic pathways and the resultant proteotoxic stresses are considered a secondary hallmark of various types of malignancies, including BCs [117]. Many approaches have been made in treating BCs by targeting the protein quality control pathways [118]. Several BC cells have been shown to overexpress and/or amplify LON protease, which is connected with mitochondrial proteome dysfunction, metabolic reprogramming, and determining the effectiveness of chemotherapy [119,120]. Thus, StAR, as a regulator of LON-mediated protein quality control processes, may trigger the growth of luminal subtype BCs. 

## 8. Therapeutic Strategies for the Treatment and Management of BCs

Therapeutic approaches for the treatment and management of BCs are largely dependent on type and stage of tumors, overall health of patients, ER, PR, and HER2 expression levels, and gene expression/mutation profiles, as well as metastasis of cancerous cells. Patients afflicted with ER+/PR+ breast tumors are candidates for hormone therapies and have a better chance of survival. Conversely, hormone-independent tumors do not respond to endocrine therapies and have a higher tendency of cancer recurrence [121]. In localized BCs, surgical removal of tumors is the primary treatment of choice in managing the disease. In metastatic BCs, debulking the tumor to an appreciable size followed by chemotherapy is a better option for higher success. In addition to surgical interventions, other factors, such as low disease burden and metastasis to soft and bone tissues, are associated with long-term survival of patients with metastatic HER2+ BC treated with HER2 targeted therapies [122]. 

Treatments with platinum-based drugs have shown to have beneficial effects in patients with metastatic TNBCs. These drugs act on cancer cells by forming highly reactive platinum complexes that interact and crosslink DNA molecules, thereby preventing the proliferation of malignant cells. A phase II clinical trial of platinum single-agent in metastatic TNBCs demonstrated an overall response rate of 27%. However, when the cohort was stratified based on the presence of breast cancer gene 1 and 2 (BRCA1 and BRCA2) mutations, there was an increase in response rate (55%) in patients with the mutations when matched with response rate of 20% in those with wild-type BRCA1 and BCAR2 genes [123]. Nonetheless, an earlier clinical trial showed a higher overall survival rate (~80%) in metastatic TNBC patients with BRCA1 &BRCA2 mutations who received cisplatin monotherapy [124]. Chemotherapeutic agents, including anthracyclines and taxanes, are frequently used for treating patients with TNBCs [125,126]. A study conducted to assess the efficacy of taxanes on metastatic BC patients carrying BRCA1 and BRCA2 revealed that these mutations with hormone-independent status were less responsive to taxanes [127]. Despite these therapeutic opportunities, the increasing occurrence of drug resistance and toxicity has necessitated the development of other antitumor agents. Alternative treatments available for mitigating drug resistance include monoclonal antibodies, check point inhibitors, immune modulators, vaccines, adoptive cell transfer, oncolytic viruses, and adjuvant immunotherapies [128,129]. Humanized monoclonal antibodies, such as trastuzumab and pertuzumab, directed against *HER2* have proven to be good chemotherapeutic candidates in treating HER2+ BCs [130]. However, reports on trastuzumab resistance have been on the rise and warranted development of new HER2 targeted interventions [131]. An improved programmed cell death by H2-18, an antibody raised against HER2, resulted in a marked anti-tumor activity on trastuzumab-resistant BC cells. Many combination studies of trastuzumab with other chemotherapeutic agents are emerging to combat metastatic HER2+ BCs with favorable outcomes [132,133,134]. Additionally, HER2 derived peptides, such as E75 and GP2, have been targeted for vaccine development for certain BCs to elicit tumor-specific immune responses [135]. 

ERs have largely been the target of hormone therapy for the management of hormone-sensitive malignant breast tumors. Tamoxifen, a first generation SERM, is the standard adjuvant endocrine therapy in post-menopausal women who are unable to take AIs because of adverse side effects [136]. Tamoxifen was effective in preventing estrogen stimulated cell proliferation in BCs; however, it acts as an agonist on endometrium, ovary, and uterus [2]. Subsequently, second (raloxifene) and third (lasofoxifene) SERMs have been developed; the latter has been demonstrated to be most potent for the prevention and treatment of BCs, resulting in an 83% decrease in ER+ breast cancer incidence [137]. As stated above, SERMs and/or tamoxifen, by inhibiting cholesterol and its specific metabolites, and LXR events, play vital roles in preventing ER+/PR+ BCs. Alternatively, a selective estrogen downregulator, fulvestrant, a 7*α*-alkylsulphinyl analogue of E2, binds ER with high affinity and competitively inhibits binding of E2 to ERs. Fulvestrant prevents the dimerization of ERs and transcriptional activation of ER-related genes [138]. It has been shown that fulvestrant decreases tumor progression in TNBCs, and in a combination therapy with tamoxifen inhibits the proliferation of ER+ BCs by upregulating ERβ expression [139].

Considering the importance of aromatase in estrogen biosynthesis, AIs have been frequently used for combating ER+/PR+ BCs. AIs are grouped into first, second, and third generation based on their chemical structure and impact in inhibiting aromatase activity involving E2 biosynthesis, and are categorized as steroidal and non-steroidal, and reversible and non-reversible inhibitors [140]. Aminoglutethimide, a first-generation AI, lacks the ability to selectively target aromatase and indiscriminately inhibit non-target cytochrome P450 enzymes, thereby necessitating glucocorticoid augmentation. The use of aminoglutithemide has also been reported to cause adverse side effects, such as lethargy, rashes, anemia, nausea, and fever. Because of non-specificity and unfavorable reactions to aminoglutethimide, second generation AIs have been developed, which include fadrozole, vorozole, and formestane, which are not very specific to aromatase. Later, third generation Ais, i.e., anastrozole, letrozole, and exemestane, have been shown to be highly potent and selective for aromatase. Formestane and exemestane are steroidal AIs derived from androstenedione, and they inhibit E2 synthesis by acting as a pseudo substrate, binding irreversibly to the androgen-binding domain of aromatase [2,78]. Conversely, non-steroidal AIs, such as aminogluthemide, fadrozole, vorozole, anastrozole, and letrozole, function by interacting with aromatase in a reversible manner. Despite the efficacy of third generation AIs in the treatment of hormone-sensitive BCs with moderate side effects, resistance to AI has been attributed to the overexpression of HRs, as well as the induction of growth factor receptor signaling pathways, all of which constitute potential targets for chemotherapeutic improvements in managing AI-resistant BCs [141]. Generally, the third-generation AIs are well tolerated and more effective than those of tamoxifen and SERMs, and have been widely used for the management of HR-positive BCs in post-menopausal women. During AI therapy, BC cells undergo continued deprivation of estrogen along with disruption of ER signaling. This, in turn, induces cells to develop ER sensitivity, adopt changes, and evolve alternate pathways, including MAPK, PI3K, and mTOR [142]. Although considerable success has been achieved with third generation AIs with minimal side effects in post-menopausal women, resistance to endocrine therapy is critical with increased tumor progression, acquisition of malignant phenotypes, and poor prognosis. Despite the effectiveness of endocrine therapies, they diminish estrogen levels throughout the body, thus, generating undesirable side effects, including osteoporosis, breast atrophy, vaginal dryness and atrophy, depression, reduced libido, and carcinogenesis to other tissues in certain cases [2,21,143]. 

## 9. Impact of HDACIs in BCs: Possible Alternatives to Endocrine Therapies

The HDAC family enzymes (epigenetic regulators) are frequently altered, dysregulated, and mutated in BCs and many other cancers. HDACs regulate not only the acetylation of histones in nucleosomes, but also a variety of non-histone substrates, including many proteins that are involved in tumorigenesis, angiogenesis, apoptosis, and cell invasion [144]. Subsequently, HDACIs have multiple targets on diseases/cancers, and have been demonstrated to produce favorable outcomes on various aspects, such as cell cycle arrest, anti-proliferation, apoptosis, differentiation, senescence, anti-angiogenesis, and autophagy [27,145]. It has been reported that Sirt1, a class III HDACI, regulates the expression and activity of aromatase in BCs [21,146]. Furthermore, inhibitors of Sirt 1/2, cambinol and inhibitor VII, have been shown to decrease aromatase mRNA and protein levels in ER+ (MCF7) and ER- (MB-231) BC cell models [146]. This is especially important since aberrant aromatase expression has been implicated in intra-tumoral estrogen production. Due to varied effects of HDACs, considerable attention has been placed on the development of HDACIs for the treatment of BCs. These inhibitors are widely used as anti-cancer drugs, especially for refractory cutaneous and peripheral T cell lymphoma, and multiple melanoma [147]. A number of HDACIs, including valproic acid, entinostat (MS-275), MGCD0103, and PCI-24781 (Abexinostat), are also in various phases of clinical trials against breast malignancies. However, HDACIs, either alone or in combination with other anti-cancer drugs, are clinically efficacious, robust, and safe, and display limited toxicity (in comparison to AIs) against multiple oncogenic events [148].

Studies have demonstrated the anti-tumorigenic effect of HDACIs in various malignant diseases, including BCs, and their ability to reverse endocrine therapy resistance in metastatic BC patients has also been reported [149,150]. Specifically, vorinostat can cross the blood-brain barrier and has been shown to prevent ~62% brain metastases by inducing DNA double-strand breaks in TNBCs [26]. In an in vivo mouse model of metastatic BC (4T1 mouse TNBC cells), vorinostat has been demonstrated to suppress lung metastasis [151]. Panobinostat, an HDAC6 inhibitor, has been reported to enhance acetylation of H3 and H4 histones, induce apoptosis, decrease proliferation, and reactivate *ER mRNA* and protein expression in TNBC cells [152]. Panobinostat is regarded as an inhibitory modulator of aromatase, either alone or in combination with letrozole, in human MCF7/H295R co-cultures [152,153]. In line with these findings, we provided evidence that a number of HDACIs, by altering StAR acetylation patterns, are capable of decreasing StAR, aromatase, and E2 levels in ER+/PR+ BC cells [23]. Nevertheless, panobinostat can overcome AI resistance by suppressing the level of NF-kB1 in AI-resistant BC cells [154]. Both vorinostat and panobinostat have been shown to acetylate heat shock protein 90 (Hsp90) and degrade Hsp90 client proteins, such as ErbB2 and HDAC6 in ER-BCs [155,156]. The inhibition of HDACs has been shown to promote ubiquitin-dependent proteasomal degradation of DNA methyltransferase 1 in BC cells [157,158]. It has been reported that romidepsin induces acetylation of H3 and apoptosis, and subsequently suppresses vascular epithelial growth factor and HIP1α [159]. Both trichostatin A (TSA) and belinostat, in combination, have been demonstrated to degrade ERα and altered cell proliferation in MCF7 cells, in which belinostat also induces histone H4 acetylation [160].

HDACIs of different classes are in various phases of BC clinical trials. Entinostat has been reported to inhibit tumor initiating cells (TIC) that are linked to metastasis and drug resistance. This inhibitor also reverses the epithelial-to-mesenchymal transition and reduces TIC cell proliferation in TNBCs [161]. In a phase II study, the addition of entinostat to exemestane treatment for advanced ER+ BCs showed an increase in overall survival rate, when compared with exemestane plus placebo treatment. Entinostat was also capable of decreasing the number of myeloid-derived suppressor cells (MDSC), and downregulated CD40 expression. This suggest a mechanism that includes immune modulation, since MDSCs play pivotal roles in the immune escape of malignant cells by stimulating regulatory T-cell proliferation and cytokine secretion by type 1 helper T-cells [162,163].

Valproic acid (VPA), widely used to treat neurological diseases, such as epilepsy and bipolar disorder, has been shown to inhibit cell proliferation and induce apoptosis in HER2+ SKBR3 BC cells. Furthermore, VPA downregulates pro-survival protein Bcl-2 and induces apoptosis in MCF7 cells [164,165]. TSA has been demonstrated to display antineoplastic activity in vitro and in vivo in BC cell models, in which TSA inhibits cell proliferation by mediating H4 hyperacetylation. The anti-tumorigenic activity of TSA has also been reported in rat mammary carcinoma cells [166,167]. Anti-proliferative effects of TSA are associated with the increased expression of 15-Lipoxygenase-1, a key enzyme mediating oxidative metabolism of polyunsaturated fatty acids, which is accompanied by cell cycle arrest and apoptosis [168,169]. TSA is also reported to enhance the acetylation and stability of ERα involving the p300 protein which blocks ERα ubiquitination [170]. Another novel class I/II HDACI, CUDC-907, affects PI3K in advanced/relapsed myeloma in a phase I trial [171].

The efficacy and specificity of a number of HDACIs, in clinical trials, in reversing resistance and re-sensitizing breast tumors are promising (Table 2). For example, in a phase II trial, 43 patients with ER+ metastatic BCs were treated with a combination of tamoxifen and vorinostat, which showed an over 40% reduction in tumors. Follow up studies employing tamoxifen-resistant ER+ BCs have identified the increased expression of both pro-survival protein Bcl-2 and ER as key drivers of hormone resistance. The treatment of ER+ BCs with the HDAC inhibitor PCI-24781 re-sensitized tumors to tamoxifen, resulting in the downregulation of Bcl-2, leading to apoptosis [172,173]. It has been demonstrated that both PCI-24781 and ER inhibition downregulate AKT, and simultaneously induce BC cell deaths [159]. The treatment of TNBC cells with vorinostat and FTY720 (fingolimod, a multiple sclerosis drug and a class I HDACI) has been shown to reactivate the expression and function of ERs [174].

Chemotherapy drug resistance is a major obstacle in the effective treatment of metastatic and/or aggressive BCs. Increasing evidence indicates that HDACIs reverse resistance to chemotherapy drugs, thus, restoring their efficacy for treating relevant diseases. Paclitaxel, a first-line chemotherapy drug for metastatic BCs, blocks the progression of mitosis by stabilizing the microtubule polymer [189]. The use of HDACIs has been shown to reverse paclitaxel resistance in MCF7 cells. Treatment with an HDACI, salvianolic acid A, has been shown to exhibit an inhibitory effect on the PI3K/Akt pathway and reverse paclitaxel resistance [28,167] The synthetic HDACI, FA17, has been reported to reverse AI resistance in MCF7 cells [190]. Radiation therapy plays an important role in the treatment of metastatic BCs, especially in patients with a high risk of locoregional recurrence following mastectomy. Since the majority of BC patients who undergo lumpectomy receive radiation therapy, the effectiveness of this treatment can impact disease control as well as overall survival [191]. In this context, vorinostat enhances the radio sensitivity of tumor tissue and increases the efficacy of radiation therapy in TNBC cells [151,192]. As well, VPA has been demonstrated to increase the radio sensitivity in MCF7 cells through the accumulation of DNA strand breaks [193]. Based on clinical and experimental evidence from various reports, including our own, regarding the effects of HDACIs and other anti-cancer drugs, a schematic model illustrating mechanistic insights into therapeutic strategies for the prevention and treatment of BCs is proposed (Figure 7). 

## 10. Conclusions and Perspectives

Estrogens, especially E2, play crucial roles in the progression and survival of hormone-sensitive BCs, the most prevalent non-cutaneous cancer in women globally. Aromatase catalyzes the final step in estrogen biosynthesis, i.e., aromatization of androgens to estrogens, and it is highly expressed in both non-cancerous and cancerous breast tissues, suggesting its importance in both breast physiology and pathophysiology [23,24,25]. The ovary, especially its granulosa cells, is the major source of estrogens in pre-menopausal women; however, extra-ovarian sites synthesize large amounts estrogens in post-menopausal women. Hormone-sensitive BCs are ER+/ERα, PR+, HER2/ErbB2+, or all three, in which E2 is critical for cancer growth, survival, and maintenance. Whereas therapeutic strategies for hormone dependent BCs involve the selective suppression of ER by either anti-estrogens or AIs in post-menopausal women, they display adverse side effects [5,9,10]. Despite the effectiveness of AIs in the management of recurrent and/or metastatic BCs, resistance to endocrine therapy is the major cause of cancer death. To alleviate underlying AI resistance, novel approaches in combating carcinogenic events, with increased specificity and efficacy, are warranted. Technological advances have targeted therapeutic strategies with markers of epigenetic regulators since the dysregulation of HDACs is a primary event in BC tumorigenesis [145,194]. The inhibition of HDACs results in the acetylation of numerous histone and non-histone substrates, including tumor suppressor proteins and oncogenes [195]. HDACIs, possessing anti-tumorigenic activities, represent a novel category of drugs for the treatment of BCs, and many non-malignant and malignant diseases [148]. 

In consideration of the advancement of HDACIs in combating breast carcinogenesis with promising outcomes, our data provided evidence that a number of HDACIs, at doses frequently utilized in the clinics, by modulating StAR acetylation patterns, decreases StAR and E2 levels in ER+/PR+ BC cells [23], suggesting that StAR could be targeted in the management of hormone-sensitive BCs. Whereas the involvement of StAR in the growth and progression of ER+/PR+ BCs remains elusive, analyses of TCGA/cBioPortal research datasets for genomic and molecular profiles of steroidogenic enzymes, demonstrated amplification of the *StAR* gene in poor survival of patients afflicted with luminal subtype BCs, providing insight into a vital influence of StAR on this deadly disease [2,24]. Nonetheless, we reported that StAR is abundantly expressed, concomitant with E2 synthesis, in hormone-sensitive BC cells, in comparison with little to none in either non-cancerous mammary epithelial cells or TNBCs [23]. Consistent with overexpression of StAR in ER+/PR+ BCs, higher levels of StAR were observed in three mouse models of breast tumors (induced by mouse mammary tumor virus promoter driven *cNeu* and *H-Ras* oncogenes, and polyomavirus), when compared with respective genetic background transgenic mouse mammary tissues (Manna PR et al., manuscript in preparation). Strikingly, the abundant expression of StAR, in hormone-sensitive BC cell lines and mouse models, is coordinately associated with E2 levels, pointing to the involvement of StAR in the biologic behavior and/or pathogenesis of hormone-sensitive BCs. 

Taken together, the aberrant high expression of StAR facilitates abnormal delivery of cholesterol to the inner mitochondrial membrane, resulting in an adequate availability of precursors for estrogen/E2 biosynthesis in promoting breast tumorigenesis, which could be a plausible mechanism in the development and maintenance of ER+/PR+ breast malignancies. Additionally, the profound expression of StAR in hormone-sensitive BCs, in comparison with their normal counterparts or TNBCs, demonstrated that StAR can serve as a novel prognostic marker in luminal subtype BCs. Considering the heterogeneous nature of BCs, the combinatorial therapy of HDACIs with AIs, markers of DNA-damaged genes, or other anti-cancer drugs, acting selectively and effectively on multiple targets, would be a rational approach in the prevention and/or treatment of hormone-sensitive BCs or other relevant cancers.

## Figures and Tables

**Figure 1 biomedicines-10-01313-f001:**
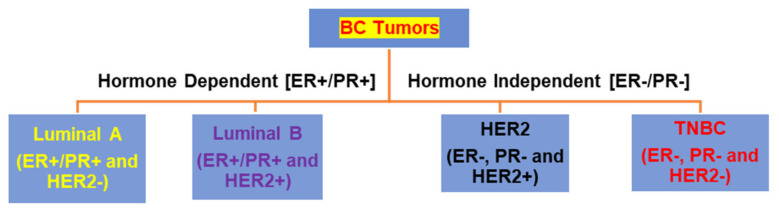
BC tumors have traditionally been classified into four subtypes: Luminal A, Luminal B, HER2, and TNBC. Classification of different tumor subtypes is based on IHC markers for ER, PR, and HER2 expression which are represented by either positive (+) or negative (−) signs.

**Figure 2 biomedicines-10-01313-f002:**
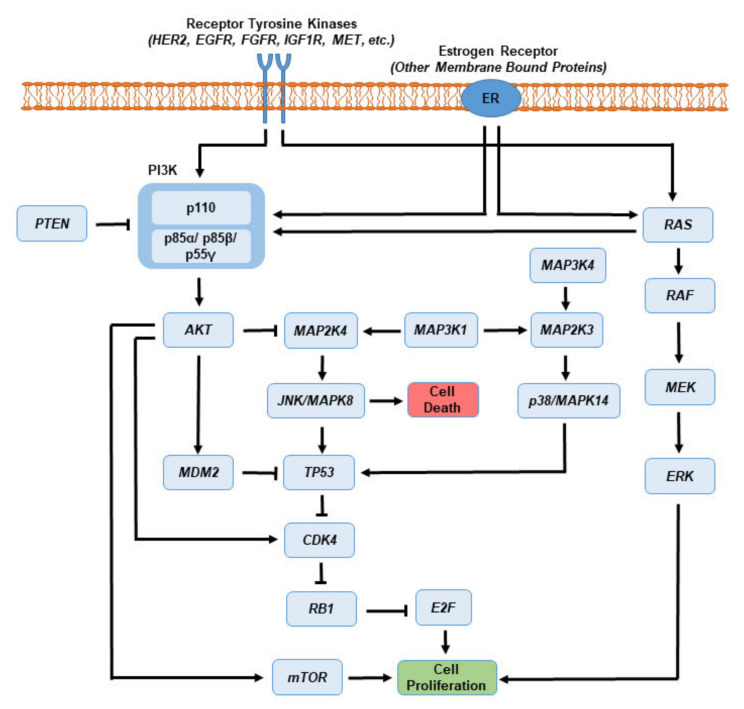
Involvement of frequently altered pathways in BCs include PI3K/AKT, KRAS/RAF/ERK, and TP53/RB1. Mutations of these signaling have been identified in genomic studies of different types of BC tumors, and are believed to play a critical role in tumor proliferation and resistance to apoptosis. PI3Ks, phosphatidylinositide 3-kinases; PIK3CA, PI3K catalytic subunit α; PIK3R1, PI3K regulatory subunit α; PTEN, phosphatase and tensin homolog; AKT, protein kinase B; mTOR, mechanistic target of rapamycin; KRAS, V-Ki-ras2 kirsten rat sarcoma viral oncogene homolog; RAF, v-Raf murine sarcoma viral oncogene homolog B; MEK/MAP2K/MAP2K3/MAP2K4, mitogen-activated protein kinase kinase; MAP3K1/MAP3K4, mitogen-activated protein kinase kinase kinase; MAPK8/JNK, mitogen-activated protein kinase 8; MAPK14/p38, mitogen-activated protein kinase 14; MDM2, mouse double minute 2 homolog; TP53, tumor suppressor p53; CDK4, cyclin-dependent kinase; RB1, retinoblastoma.

**Figure 3 biomedicines-10-01313-f003:**
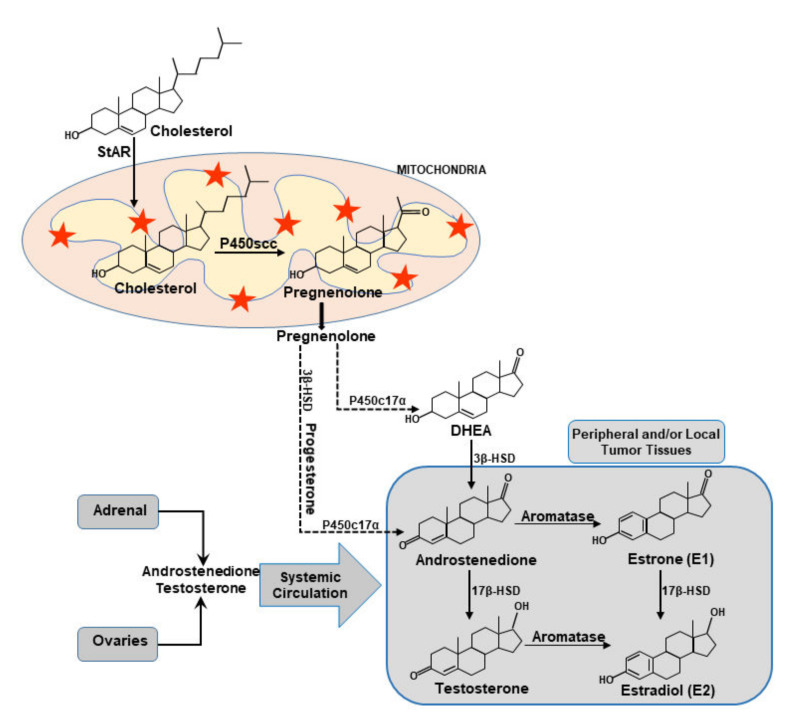
Steroid biosynthetic pathway elucidating peripheral and local tissue estrogen biosynthesis. Cholesterol is the substrate for all steroid hormones. The StAR protein predominantly regulates steroid biosynthesis by controlling the transport of cholesterol from the outer to the inner mitochondrial membrane. At the inner mitochondria, the first steroid, pregnenolone, is formed by the action of the cytochrome P450scc enzyme. Pregnenolone exits the mitochondria and it is then converted to various steroid hormones by tissue-specific enzymes. Aromatase is the rate-liming enzyme in the biosynthesis estrogens from androgens. The rectangle illustrates estrogen biosynthesis in peripheral and local tumor tissue. Androgens synthesized in the adrenal gland and ovaries (pre-menopausal) are converted to estrogens through the action of the aromatase enzyme. The 17β-HSD enzyme converts estrone to 17β-estradiol (E2).

**Figure 4 biomedicines-10-01313-f004:**
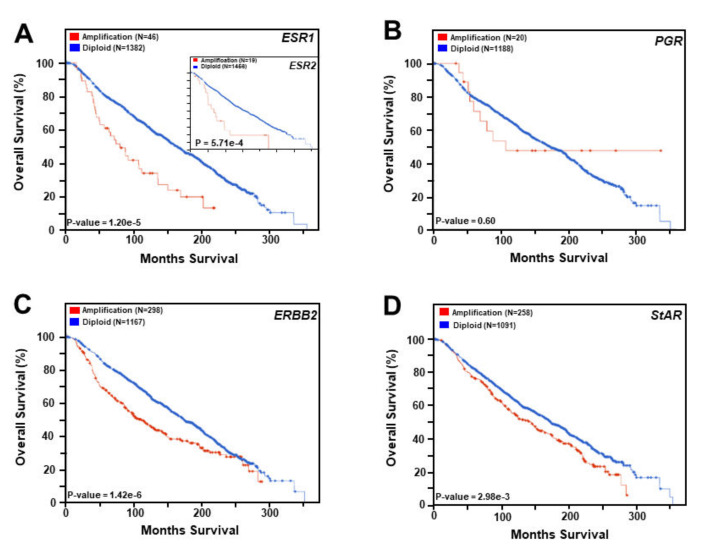
Amplification of *ESR1*, *ERS2*, *PGR*, *EBBR2*, *and StAR* genes in BC tumors and their correlation to overall survival. TCGA METABRIC breast tumor samples (N = 2173) were utilized for survival analyses. Kaplan-Meier survival curves were generated with or without amplification of the following genes: *ESR1* (**A**), *ERS2* (**A**, inset), *PGR* (**B**), *EBBR2* (**C**), and *StAR* (**D**). Overall survival rate of BC patients was assessed with ‘amplification’ and ‘diploid’ of these genes. Red and blue lines in the Kaplan-Meier plots reflect BCs with amplification and diploid status of these genes, respectively.

**Figure 5 biomedicines-10-01313-f005:**
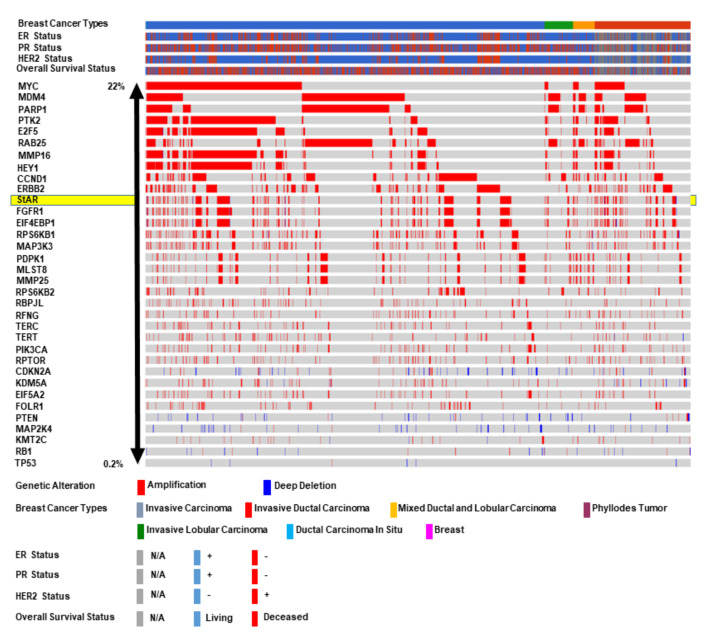
Amplification of various genes that are frequently occurred in BCs. TCGA METABRIC breast tumor samples (N = 2173) categorized in different groups (bottom panel) were analyzed for amplification of various targeted genes (illustrated at left panel). Clinical features, including ER, PR, and HER2, as well as overall survival statuses, are shown in left/bottom panels. Copy number alteration ranged from 22% (higher) to 0.2% (lower). Alteration of genes studied are the following: *MYC*, *MDM4*, *PARP1*, *PTK2*, *E2F5*, *RAB25*, *MMP16*, *HEY1*, *CCND1*, *ERBB2*, *StAR*, *FGFR1*, *EIF4EBP1*, *RPS6KB1*, *MAP3K3*, *PDPK1*, *MLST8*, *MMP25*, *RPS6KB2*, *RBPJL*, *RFNG*, *TERC*, *TERT*, *PIK3CA*, *RPTOR*, *CDKN2A*, *KDM5A*, *EIF5A2*, *PTEN*, *MAP2K4*, *KMT2C*, *RB1*, *AND TP53*. Both amplification (red) and deep deletion (blue) are highlighted.

**Figure 6 biomedicines-10-01313-f006:**
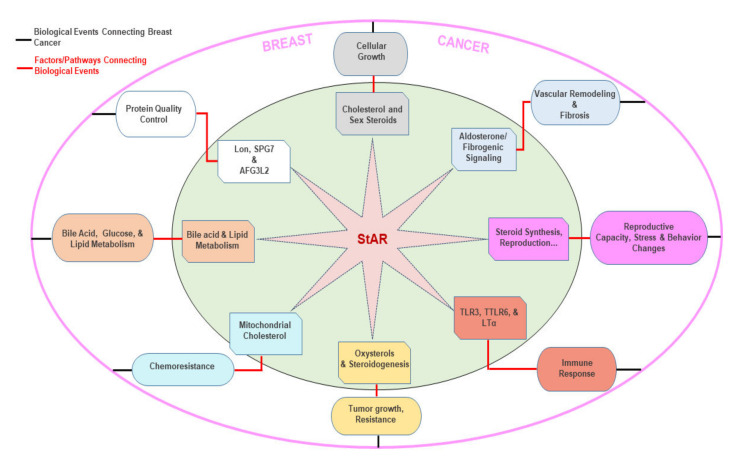
Multiple factors and/or signaling putatively influence StAR-mediated breast tumorigenesis. A hypothetical model illustrates association of a variety of factors and/or signaling pathways in growth and progression of BCs. The red line predicts factors/pathways that are connected with biological events. The black line, including StAR, with various events, predicts biological events that are connected with BCs.

**Figure 7 biomedicines-10-01313-f007:**
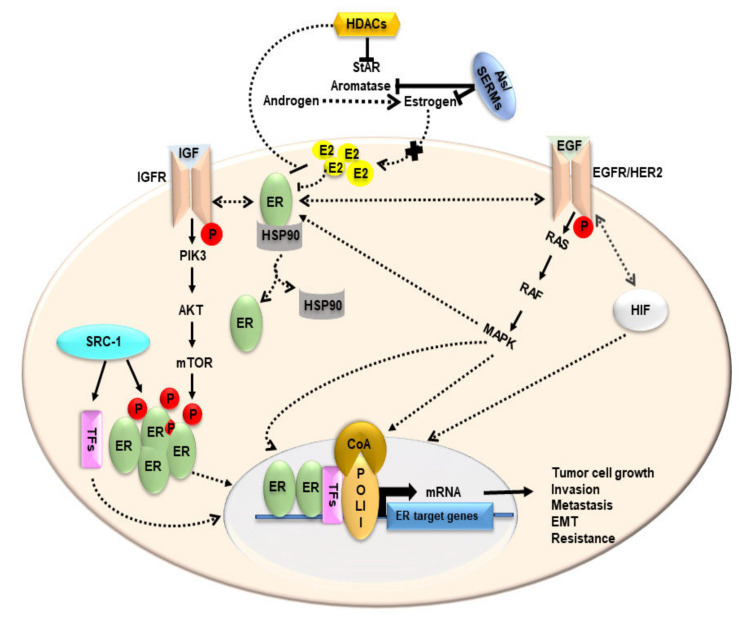
A schematic model illustrating various signaling processes in BCs. HDACIs decreases in StAR, aromatase, E2/estrogen levels. AIs inhibit aromatase and thereby block the conversion of androgens to estrogens. E2 is diminished and tumor cells engage in other non-estrogen related pathways for their growth and survival. Upon ligand activation, receptors for a number of growth factors, such as IGF and EGF, become phosphorylated (P). ER interacts with growth factor receptors (e.g., IGFR and EGFR/HER2) that trigger downstream growth signaling pathways, such as PIK3/AKT/mTOR and Ras/Raf/MAPK. This leads to phosphorylation of ER which dissociates from its chaperone HSP-90. Phosphorylated ER translocate into the nucleus to bring about transcription of ER target genes in the presence of transcriptional machinery, which consist of RNA polymerase (POL II), TFs, and coactivators (CoA). Src-1 phosphorylates ER and TFs to bring about gene transcription; whereas under hypoxic conditions, HIF mutually interacts with EGFR and results in transcriptional regulation of tumor growth-related genes. Different pathways engaged by estrogen-deprived conditions result in tumor cell growth, invasion, metastasis, epithelial to mesenchymal transition (EMT) and inhibitor resistance in BCs.

**Table 1 biomedicines-10-01313-t001:** Genomic, clinical, and pathological features of TCGA BCs classified by molecular subtypes [28].

Alterations	Luminal A	Luminal B	HER2E	Basal-Like
**ER+/HER2− (%)**	87	82	20	10
**HER2+ (%)**	7	15	68	2
**TNBCs (%)**	2	1	9	80
** *mRNA* ** **expression**	High ER, low proliferation	Lower ER, high proliferation	HER2 amplicon signature, high proliferation	Basal signature, high proliferation
** *TP53* ** **mutation**	12%	32%	75%	84%
** *MAP3K1* ** **mutation**	14%	5%	-	-
** *GATA3* ** **mutation**	14%	-	-	-
** *PIK3R1* ** **mutation**	-	-	8%	-
** *PIK3CA* ** **mutation**	49%	32%	42%	7%
** *PTEN* ** **mutation/loss**	13%	24%	19%	35%
** *INPP4B* ** **loss**	9%	16%	30%	30%
** *RB1* ** **expression**	high	-	-	low
** *RB1* ** **mutation/loss**	-	-	-	20%
**Cyclin D1 amplification**	29%	58%	38%	-
**Cyclin E1 amplification**	-	-	-	9%
** *CDK4* ** **gain**	14%	25%	24%	-
**Grade**	Low	Moderate	High	High
**Prognosis**	Good	Intermediate	Poor	Poor
**Targeted Therapies**	Endocrine		HER2 targeted therapy (e.g., Trastuzumab)	Chemotherapy, Investigational

**Table 2 biomedicines-10-01313-t002:** A variety of HDACIs at different phases of BC clinical trials.

HDACIs	Targets	Phase(s)	Type of Therapy	BC Subtypes	References
Vorinostat (SAHA)	Class I, II, IV	II	Combination	ER+/PR+	[172]
		I/II	Combination	HER2 amplified	[175]
		I/II	Combination	Metastatic or recurrent	[176]
		II	Combination	HER2+	[177]
		I/II	Combination	HER2+	[178]
Panobinostat (LBH589)	Class I, II, IV	I	Combination	TNBC	[179]
		I/II	Combination	TNBC	[150]
		I	Combination	Metastatic or recurrent	[180]
Entinostat (MS275)	Class I	I	Single	Metastatic/unresectable with no effective treatment	[181]
		II	Combination	ER+, relapsed/progressed	[162,163]
		III	Combination	ER+/PR+, metastatic/locally advanced	[182]
		II	Combination	TNBC, metastatic/locally advanced	[183]
		I	Combination	HER2-, metastatic/locally advanced	[184]
		I	Combination	HER2+, metastatic or recurrent	[185]
Romidepsin	Class I	I/II	Combination	TNBC or BRCA1/BRCA2, metastatic/locally recurrent	[186]
Valproic Acid	Class I, II	I	Combination	Metastatic/locally advanced	[172]
Rocilinostat (ACY-1215)	Class II	I	Combination	Metastatic/unresectable with no effective treatment	[187]
CUDC-907	Class I, II	I	Single	ER+/PR+, HER2-, advanced	[188]

## Data Availability

Not applicable.

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
