# Peer review of "Hormonal and Genetic Regulatory Events in Breast Cancer and Its Therapeutics: Importance of the Steroidogenic Acute Regulatory Protein"

_biomedicines, 2022, doi:10.3390/biomedicines10061313_

Round 1

Reviewer 1 Report

This is an in silico study that requires experimental validations

line 87, regarding star expression and patient survival (DFMS, OS or RFS) the opposite correlation can be found (see Breast Cancer Gene-Expression Miner v4.8 (http://bcgenex.ico.unicancer.fr/) or Kmplot (https://kmplot.com/))

line 108 and 109 use ERalpha to specify the product of ESR1 gene.  ERbeta  detection is not routinely used to stratify BC

line 135-136, a reference is required.

line 140, despite these facts estrogens are known to induce apoptosis in BC cells and to overcome resistance to hormone therapy (see Jordan VC recent work)

line 253.  to date ERalpha is the only validated marker of estrogen mitogenic effects

off-targets known to mediate some antiestrogens pharmacological effects has to be considered, because they have been linked to tumor escape to treatments

in particular SERMS were shown to affect cholesterol metabolism at different levels which may affect the dynamic of cholesterol metabolism in BC cells

the authors over-referred to reviews rather than original research studies.

Author Response

Response to Reviewer 1 Comments

Point 1: This is an in silico study that requires experimental validations

Response 1: Please note that this is a review article; however, in-silico data involving genomic studies with StAR, (and steroidogenic enzymes), a new entity in breast cancers (BCs) and the major theme of this manuscript, have been strengthened with experimental validations (Manna PR et. al., 2016, Prog Mol Biol Transl Sci.; 2019, Cancers, and 2019 Biochem Biophys Res Commun.;). Herein, we have summarized the current understanding of hormonal and genetic regulatory events that drive breast tumorigenesis, especially the most prevalent hormone sensitive BCs. We were the first to report, upon analyses of The Cancer Genome Atlas (TCGA) and cBioPortal research datasets, that amplification of the StAR gene, but not other steroidogenic enzyme genes, including aromatase, was correlated with poor survival of patients afflicted with BCs (Manna PR et. al., 2016, Prog Mol Biol Transl Sci., and 2019, Cancers). Accordingly, we reported that the StAR protein is abundantly expressed in hormone receptor positive (ER+/PR+) BCs, (concomitant with E2 accumulation), in comparison with little to none in their non-cancerous counterparts (Manna PR et. al., 2019, Biochem Biophys Res Commun). Concurrently, a post-translational modification (PTM) of StAR, acetylation, was discovered, in which 11 StAR acetyl lysine residues, under basal and in response to histone deacetylase inhibitors (HDACis), were identified by LC-MS/MS. The involvement of StAR, in luminal subtype BCs, is verified in mouse models of breast tumors induced by MMTV promoter driven cNeu and H-Ras oncogenes, and polyomavirus, as indicated in the manuscript (Manna PR et al, manuscript in preparation). To our knowledge, this is the first review article that opens up a new avenue in BC research demonstrating the following novel insights: i) differential expression of StAR in non-cancerous vs. cancerous breast tissues indicates that StAR can be considered as a diagnostic and/or prognostic marker in BCs, ii) StAR is identified as a novel acetylated protein implicating a potential new mechanism in steroid biosynthesis, iii) depletion of StAR is coordinately linked with repression of E2 accumulation, signifying therapeutic potentials of StAR in ER+/PR+ BCs, iv) aromatase, the key enzyme in estrogen synthesis, is highly overexpressed not only in BCs, but also in non-cancerous mammary epithelial cells, suggesting the involvement of additional factor(s) in BCs, in which StAR fulfills all relevant criteria to become a novel candidate, v) regulation of E2 synthesis is associated with StAR, but not aromatase, pointing to an indispensable role of StAR in the pathogenesis of hormone sensitive BCs, and vi) HDACis, a novel category of anti-cancer drugs, by altering StAR acetylation patterns, decreases E2 synthesis, thus, these epigenetic regulators could be targeted as alternatives of endocrine therapies (that are allied with undesirable side effects) for the prevention and treatment of the majority of hormone dependent BCs (>80% of all BCs). We sincerely hope that the reviewer will agree with us.

Point 2: line 87, regarding star expression and patient survival (DFMS, OS or RFS) the opposite correlation can be found (see Breast Cancer Gene-Expression Miner v4.8 (http://bcgenex.ico.unicancer.fr/) or Kmplot (https://kmplot.com/))

Response 2: The statement made on line 87 sentence was based on analyses of a number of TCGA research datasets, in which amplification of the StAR gene (observed between 12 and 26%) was correlated with poor survival of BC patients (Manna PR et. al., 2016, Prog Mol Biol Transl Sci., and 2019, Cancers). Specifically, genomic profiling of breast tumors available in different TCGA/cBioPortal studies/publications were analyzed for DNA copy number alterations (CNAs) for StAR and steroidogenic enzyme genes using the GISTIC 2.0. algorithm, in which CNA data were categorized as high-level amplification (+2 copies), gain (+1 copy), diploid (normal/no change), homozygous deletion (-2 copies), and hemizygous deletion (-1 copy). StAR CNA data were further verified by Spearman’s rank coefficient analysis for their correlation to StAR mRNA with RNA-Seq data, using the RSEM algorithm (Manna PR et. al., 2019, Cancers). Regarding overall survival (OS) analyses, we generated Kaplan-Meier survival curves, connecting different TCGA/cBioPortal data cohort, using with (as high-level Amplification) and without (Diploid) amplification, which demonstrated that amplification of the StAR gene significantly affected OS of BC patients (Manna PR et. al., 2019, Cancers). Furthermore, analyses of tumors (T), nodal (N) status, and metastases (M) of BC tumors expressing StAR mRNA resulted in cancer deaths in stage specific manners in different TNM categories. Collectively, genomic and molecular profiles of in-silico data were confirmed with a variety of experiments, providing novel insights that StAR plays an important role in the biologic behavior and/or pathogenesis of ER+/PR+ BCs. Currently, there is no published evidence available revealing the opposite correlation between amplification of the StAR gene (CNA data assessing kmplot with amplification vs. diploid) and OS of BC patients.   

Point 3: line 108 and 109 use ERalpha to specify the product of ESR1 gene.  ERbeta  detection is not routinely used to stratify BC

Response 3: The revision has been made according to the suggestions of the reviewer.

Point 4: line 135-136, a reference is required.

Response 4: Two references have been added in the revised version.

Point 5: line 140, despite these facts estrogens are known to induce apoptosis in BC cells and to overcome resistance to hormone therapy (see Jordan VC recent work)

Response 5: A sentence has been added in the revised version (Section 3. Molecular Pathogenesis of BCs and Frequently Altered Pathways) citing Jordan VC’s recent studies. “Noteworthy, however, estrogens are known to induce apoptosis in BC cells and are capable to overcome endocrine therapy resistance {Maximov, 2020 #6739;Abderrahman, 2020 #6738}.”  

Point 6: line 253.  to date ERalpha is the only validated marker of estrogen mitogenic effects

Response 6: The sentence associated with ‘line 253’ has been revised to specify the information.

Point 7: off-targets known to mediate some antiestrogens pharmacological effects has to be considered, because they have been linked to tumor escape to treatments

Response 7: We agree with the reviewer regarding the well-known off-target effects of antiestrogens/(endocrine therapies), even though they have been frequently used in the clinics for the treatment of ER+/PR+ BCs in post-menopausal women. As such, a brief information describing undesirable side effects of endocrine therapies such as osteoporosis, vaginal dryness and atrophy, depression and reduced libido, and carcinogenic to other tissues, has been added under Section 8 (Therapeutic Strategies for the Treatment and Management of BCs).

Point 8: in particular SERMS were shown to affect cholesterol metabolism at different levels which may affect the dynamic of cholesterol metabolism in BC cells

Response 8: Effects of SERMs on cholesterol trafficking and metabolism, with reference to BC cells, have been discussed in respective places (sections 4 and 7) in the revised version of the manuscript.

Point 9: the authors over-referred to reviews rather than original research studies.

Response 9: Review articles are preferably cited, as they generally provide most comprehensive information that originates from a number of original research studies included in a particular review. Therefore, we believe that reviews, including original research publications, are beneficial for enthusiastic and vivid reader/scientist. While we have cited original research articles in relevant places, a few review articles are appropriately replaced in the revised version, as concerned by the reviewer.    

Reviewer 2 Report

Dear Authors,

I congratulate you on the very interesting topic. Your paper is a well written, detailed and comprehenseve review about the hormonal, molecular and genetic factors that are involved in the development and progression of breast cancer (BC). I really appreciate the evidence provided about the role of StAR protein as a key element in the regulation of steroid biosynthesis and also the evaluation of possible alternatives to standard endocrine therapies provided by HDAC inhibitors, suggesting that StAR could be targeted in the management of hormone sensitive BCs and could be considered as a novel prognostic marker in luminal subtypes BCs.

Given these consideration, I think that your paper could be accepted for publication after these simple minor corrections:

  • A minor spell check; please, also, check that all the abbreviations have been correctly used and consider to add a list of abbreviations at the end of the paper;
  • In the figure 1, I think it could be useful to change the color of the words inside the HER-2 box, avoiding the use of light blue inside a similar blue box;
  • Please, modify the table 1 by merging the cells of the headers inside the table (f. e. RB1 pathway) so that they could be easily distinguished from the genetic alterations and consider to add a null value or a slash </> or a hyphen/dash <-> in the cells without values.

Kind Regards

Author Response

Response to Reviewer 2 Comments

Point 1: Dear Authors,

I congratulate you on the very interesting topic. Your paper is a well written, detailed and comprehenseve review about the hormonal, molecular and genetic factors that are involved in the development and progression of breast cancer (BC). I really appreciate the evidence provided about the role of StAR protein as a key element in the regulation of steroid biosynthesis and also the evaluation of possible alternatives to standard endocrine therapies provided by HDAC inhibitors, suggesting that StAR could be targeted in the management of hormone sensitive BCs and could be considered as a novel prognostic marker in luminal subtypes BCs.

Given these consideration, I think that your paper could be accepted for publication after these simple minor corrections:

Response 1: We would like to thank the reviewer for constructive comments on various aspects of our manuscript and its provisional acceptance upon minor corrections. We have addressed/revised minor issues, as suggested by this reviewer, in respective places in revised version of the manuscript.

Point 2: A minor spell check; please, also, check that all the abbreviations have been correctly used and consider to add a list of abbreviations at the end of the paper;

Response 2: We thoroughly check spelling errors, and verified all the abbreviations used in the manuscript. We are also including a list of frequently used abbreviations at the end of the manuscript, as suggested by the reviewer.

Point 3: In the figure 1, I think it could be useful to change the color of the words inside the HER-2 box, avoiding the use of light blue inside a similar blue box;

Response 3: Changed the color of words for better visibility, as suggested by the reviewer.

Point 4: Please, modify the table 1 by merging the cells of the headers inside the table (f. e. RB1 pathway) so that they could be easily distinguished from the genetic alterations and consider to add a null value or a slash </> or a hyphen/dash <-> in the cells without values. Kind Regards

Response 4: We agree with the reviewer, and Table 1 has been redesigned in the revised version for better clarification. Specifically, both PIK3CA/PTEN and RB1 pathway columns are omitted, and a ‘-‘ is used when the information on a particular genetic alteration is lacking in different categories.

Round 2

Reviewer 1 Report

The authors have seriously improved their manuscript but failed to imrpove the relationship between  cholesterol and SERM that were established. 5,6-epoxycholesterol have been reported to act as signaling molecules in BC cells treated with SERMs, and more recently  oncosterone (OCDO) was reported to be produce by BC cells and to act as tumor promoter via GR, while also interacting with LXR.

Thus not considering this pahtway controled by the inhibition of the cholesterol-5,6-epoxide-hydrolase by SERMs will deprive readers from important mechanism that are under investigation for new therapeutic strategies to treat ER+ and ER- BC

Author Response

Point 1: The authors have seriously improved their manuscript but failed to improve the relationship between cholesterol and SERM that were established. 5,6-epoxycholesterol have been reported to act as signaling molecules in BC cells treated with SERMs, and more recently oncosterone (OCDO) was reported to be produce by BC cells and to act as tumor promoter via GR, while also interacting with LXR. Thus, not considering this pathway controlled by the inhibition of the cholesterol-5,6-epoxide-hydrolase by SERMs will deprive readers from important mechanism that are under investigation for new therapeutic strategies to treat ER+ and ER- BC.

Response 1: The major theme of this review article is to summarize the current understanding of hormonal and genetic regulatory events in breast cancer (BC) and its therapeutics with reference to StAR, a new entity in hormone sensitive BCs that we have identified lately. With regards to this reviewer’s original comments, the manuscript was revised by addressing the criticisms (see below underlined). In response to his/her additional concerns, we have added a paragraph (Page 6; lines 240-266), under Section 4, in the revised version, by briefly discussing those coordinated events associated with cholesterol and its metabolites and SERMs and their correlation to BCs with relevant citations. In addition, a sentence is added in Section 8 (Page 19; lines 579-580) mentioning that “As stated above, SERMs and/or tamoxifen, by inhibiting cholesterol and its specific metabolites, and LXR events, play vital roles in preventing ER+/PR+ BCs”.  We sincerely believe that ‘the relationship between cholesterol and SERM’ in more depth especially is beyond the scope of the present article, and hope the reviewer will agree with us. 

Point 8: in particular SERMS were shown to affect cholesterol metabolism at different levels which may affect the dynamic of cholesterol metabolism in BC cells

Response 8: Effects of SERMs on cholesterol trafficking and metabolism, with reference to BC cells, have been discussed in respective places (sections 4 and 7) in the revised version of the manuscript.

Round 3

Reviewer 1 Report

The authors addressed my concerns satisfactorily